# Design and Implementation of an Interactive System for Service Robot Control and Monitoring

**DOI:** 10.3390/s25040987

**Published:** 2025-02-07

**Authors:** Jonas Machado Santana, Bruno Duarte Silveira, Crescencio Lima, Jose Diaz-Amado, Cléia Santos Libarino, Joao E. Soares Marques, Dennis Barrios-Aranibar, Raquel E. Patiño-Escarcina

**Affiliations:** 1GIPAR Research Group, Instituto Federal da Bahia, IFBA, Vitória da Conquista, Bahia 45078-900, Brazil; jonasee2020@gmail.com (J.M.S.); bruno.eng1.eletrica@gmail.com (B.D.S.); crescencio@ifba.edu.br (C.L.); cleialibarino@ifba.edu.br (C.S.L.); erivando@ifba.edu.br (J.E.S.M.); 2Electrical and Electronic Engineering Department Universidad Católica San Pablo, UCSP, Arequipa 04001, Peru; dbarrios@ucsp.edu.pe (D.B.-A.); rpatino@ucsp.edu.pe (R.E.P.-E.)

**Keywords:** interaction, service robot, chatbot, system, recognition, robotics

## Abstract

This project aims to develop an interactive control system for an autonomous service robot using an ROS (robot operating system). The system integrates an intuitive web interface and an interactive chatbot supported by Google Gemini to enhance the control experience and personalization for the user. The methodology includes the integration of an API (application programming interface) to access a database storing user preferences, such as speed and frequent destinations. Furthermore, the system employs facial recognition, people groups’ recognition, and adaptive responses from the chatbot for autonomous navigation, ensuring a service tailored to the individual needs of each user. To validate the proposal, it was implemented on an autonomous service robot, integrated into a motorized wheelchair. Tests demonstrated that the system effectively adjusts the wheelchair’s behavior to user preferences, resulting in safer and more personalized navigation. The use of facial recognition and chatbot interaction provided more intuitive and efficient control. The developed system significantly improves the autonomy and quality of life for wheelchair users, proving to be a viable and efficient solution for autonomous and personalized control. The results indicate that integrating technologies like ROS, intuitive web interfaces, and interactive chatbots can transform the user experience of autonomous wheelchairs, better meeting the specific needs of users.

## 1. Introduction

Human–computer interaction (HCI) is based on the interaction between people and computers, considering theories and design techniques used to make a system interactive. HCI is based on knowledge of both the machine and the human side [1]. In this context, human–robot interactions (HRIs) are a growing area of interest in HCI due to the evolution of social robots. With the development of new technologies and the increasing use of robots in social environments, there is an urgent need to promote better communication between different intelligent systems and human actions [2].

HCIs are important for use with social service robots, as they allow users to interact with robots intuitively and efficiently. Through the interface, it is possible to provide feedback to the robot, control its actions, and access important information. Rocha et al. [3] highlight that a well-designed interface can help make the interaction with users more natural and enjoyable.

HRIs have seen significant advancements, driven by technological breakthroughs in artificial intelligence (AI), sensor technology, and multimodal interfaces. These developments are reshaping the way users interact with robots, emphasizing both the functionality and the user experience (UX). This section expands on recent advancements and addresses ethical considerations integral to the adoption and integration of robotic systems.

UX in HRIs is influenced by factors such as usability, accessibility, and emotional engagement. Advances in multimodal interfaces—including voice commands, touchscreens, and gesture recognition—have made interactions more intuitive and natural.

According to Farias [4], the implementation of chatbots facilitates interaction between people and automated systems, improving the efficiency and guidance of the service smoothly. Furthermore, Khanna et al. [5] adds that chatbots are communication systems that use natural language processing (NLP) resources to promote interaction between humans and machines in a manner similar to human language.

Therefore, the system integrates with an ROS Noetic, an open source framework used to write robot software. It includes libraries and other tools that help simplify various aspects of robot software development, as described by Quigley et al. [6]. This allows the creation of topics that interact within the system through the Roslibjs library.

To communicate the web interface with the chatbot and the database, an application programming interface (API) can be used, which is a set of rules, protocols, and tools that allow different software to communicate with each other, as explained by Xavier [7].

According to Wiehl [8] and O’Flaherty [9], the development of facial recognition technologies is based on the principle that the unique arrangement of facial features creates a distinct identity for each face. The distances between specific facial points, such as the depth of the eyes and the width of the nose, are measured and used by algorithms to compare faces and determine if two images belong to the same person.

Accordingly, this project aims to implement an interactive web system on a wheelchair with autonomous social navigation through class identification developed by Leite et al. [10] to assist individuals with motor limitations and promote greater autonomy and independence for users, providing a more intuitive and personalized control experience. Through this system, users can interact in a more effective and adapted manner to their needs, contributing to greater independence in their daily activities.

The remainder of this paper is organized as follows. Section 2 describes the related work. Section 3 presents the methodology used throughout the project. Section 4 discusses the results. Finally, Section 5 concludes this paper and summarizes its contributions and future work.

## 2. Related Work

Cerqueira [11] developed a web-based user interface for interactions with multiple robots. It explores the integration of the Rosbridge protocol and the Robot Web Tools community to incorporate web technologies in robotics. They used various tools, such as HTML, CSS, and JavaScript, and frameworks such as React and Angular to achieve the expected result. Cerqueira [11] used ROS topics to collect data from the robots and implemented a REST API for organizing and retrieving data from the ROS Master.

Coelho et al. [12] developed a web system for monitoring and controlling the security of a location using multiple robots. Using ROS (robot operating system) technologies, the system simulates an environment being patrolled by robots and manages communication between a web interface and robots. The project included the creation of a Python server to control security actions and a web interface built-in JavaScript, which allows us to interact with the server and visualization of the robots’ actions and the patrolled area’s map.

While [11] focuses on a web-based interface for multi-robot interactions and [12] develops a web system for security monitoring, our contribution lies in integrating diverse modalities such as web interface, chatbot with voice interactions, and facial recognition. This approach provides a more holistic and adaptable interaction mechanism compared to the predominantly single-mode interfaces in prior works.

The project developed by Correia et al. [13] consists of implementing an architecture called Erika, which integrates a chatbot to interact with the autonomous navigation of a wheelchair in a simulated environment. An API connects the chatbot to a web application that communicates with an ROS. The results show that the wheelchair can navigate while respecting social restrictions based on proxemic zones. Our system goes beyond a simulation by deploying in a real-world setting and incorporating additional modalities like the web interface and facial recognition, demonstrating a higher level of system complexity and real-world applicability.

Furthermore, when it comes to chatbots, Lukose et al. [14] explore the integration of chatbot technology in robots to enhance user engagement through real-time assistance and interactive dialogue. The study addresses the architecture, functionalities, and practical applications of integrating chatbots in robots, aiming to optimize user experience and interaction dynamics.

Andrej et al. [15] describes the development of a voice-controlled wheelchair for people with disabilities, using cloud-based voice recognition systems and edge computing to enable real-time maneuvering. Prototypes were built in three stages, from a small one to test feasibility to a full-size one equipped with distance sensors and an advanced control algorithm for semi-autonomous direction. Our system integrates REGROUP for people group recognition, allowing for more nuanced and socially aware navigation. This capability addresses the challenges of navigating dynamic social environments, which is not explicitly addressed in the cited works on voice control or teleoperation.

Sahoo et al. [16] explores advances in HRI (human–robot interaction) techniques, including multimodal interfaces, gesture recognition, and voice commands, and assesses their impacts on user experience factors such as usability, learnability, efficiency, and satisfaction. Complementing this, Su et al. [17] systematically reviews the state of the art of multimodal HRIs in its applications, including voice, image, text, eye movement, and touch.

In the field of teleoperation, we see the Lambert W function, which controls bilateral systems with time delay. This technique ensures stability and improves motion and force tracking, which contributes to the fields of wheelchair control engineering and human–robot interactions [18].

In summary, our contribution stands out by combining multimodal interactions, real-world adaptability through social navigation, and personalization to provide a more comprehensive and user-centric system. This approach addresses key research gaps, particularly in translating research advancements into practical applications for assistive technologies.

## 3. Development of the Interactive System

In this section, we describe the developed interactive system, as well as the implementation of the communication process between the web application and the chatbot, used for the interaction between the user and the wheelchair.

### 3.1. Workspace

For the development of this work, we used a motorized wheelchair with autonomous social navigation developed by Leite et al. [10], equipped with a 64GB RAM Jetson AGX Orin, a high-performance embedded system from NVIDIA. This system is widely used in AI applications, computer vision systems, and autonomous robots. In contrast, Leite et al. [10] chose to use a notebook as the processor in their study, although they used the same devices and sensors:LiDAR Slamtec RPLIDAR A2: A 2D laser mapping sensor that operates in 360 degrees with a frequency of 10 Hz and a maximum range of 12 m, used for obstacle detection and precise environment mapping.ZED 2i Stereo Camera: A stereo camera from StereoLabs that captures images and generates depth maps in real-time at 30 fps. It is essential for visual odometry, 3D mapping, and object recognition in dynamic environments. The stereo camera was used with the default factory calibration. For more information on recalibration, see the StereoLabs page [19].Samsung Galaxy Tab S6 Lite: Used for controlling the human–computer interface and equipped with a camera that operates at 30 fps, responsible for performing facial recognition of the user.24V DC Motors with Incremental Encoders: The encoders provide precise data on wheel rotation, allowing for accurate estimation of the wheelchair’s position and speed, aiding odometry control and contributing to autonomous navigation.

Figure 1 shows the hardware used in the wheelchair.

### 3.2. Integration of the Web Interface with ROS

The Rosbridge allows the connection of the interface with the ROS framework through the Rosbridge server library, which is responsible for bridging the communication between ROS and other systems, such as web pages and mobile applications as shown in Figure 2.

### 3.3. Web Interface

Moreira et al. [20] developed the initial design of the web interface using React, a JavaScript library to create user interfaces on web pages. This project connected the interface with an ROS. The interface was tested in a simulated environment in ROS-Gazebo.

Based on this initial project, we performed improvements focusing on wheelchair autonomy. We developed the battery charge indicator, an improvement to monitor the battery charge in real-time, with the percentage value obtained from a DC voltage sensor. This sensor was connected to the embedded system, which published the percentage value as a topic in the ROS.

Figure 3 presents the wheelchair control page; we added a Joystick in manual mode allowing the user to control the wheelchair with precision, similarly to the original factory control.

We added the automatic mode, controlled by directional arrows. In this control mode, by pressing once, the system interprets the command and the wheelchair follows the indicated direction until the user interrupts it by pressing the stop button, as shown in Figure 4.

Figure 5 presents the autonomous mode to allow the user to select the direction according to the map and monitor the location and route of the wheelchair in real time.

Moreover, we created a data page to visualize battery voltage and a status indicator for LIDAR sensors, ZED 2i stereo camera, and user camera, indicating if they are online or offline, as well as the images from the cameras, as shown in Figure 6.

### 3.4. Chatbot

Regarding the development of the chatbot, we used react-chatbotify, a React library for creating chatbots and implementing them in web interfaces with support for integrating Google’s AI, Gemini. It allows the modification of various parameters, such as the language of voice recognition, header color, and customization of the dialogue between the chatbot and user, among others. Algorithm 1 shows the step-by-step integration of the chatbot with autonomous social navigation with proxemic zones. Initially, if the environment map is unknown, the system generates maps using simultaneous localization and mapping (SLAM) [21]. Then, it identifies the environment using Finder’s proxemic zone, as implemented by Leite et al. [10], and recognizes the voice command through the chatbot. If the voice command includes the words “go” or “go to”, the system receives a goal from the chatbot and performs social navigation; otherwise, it requests Google Gemini.
**Algorithm 1** Chatbot, proxemic zone, and social autonomous navigation integration.1:**if** unknown map **then**2:    Generate Maps with SLAM3:**end if**4:Identify environment with proxemic zone from finder and groups’ recognition5:Identify voice command (chatbot)6:**if** voice command includes ‘go’ or ‘go to’ **then**7:    Receive a goal from chatbot8:    Make social navigation9:**else**10:    Request to Google Gemini11:**end if**

Figure 7 illustrates the chatbot operation, highlighting the interaction among its various components. On the front-end, the user utilizes the web interface to communicate with the chatbot, which incorporates text-to-speech and speech-to-text technologies to facilitate verbal interactions. The voice commands received are processed on the back-end server. When a navigation command is identified, navigation is activated. Otherwise, the command is directed to a web search using Google Gemini.

### 3.5. API

We developed an API to facilitate the communication between the database and the chatbot with the web interface. We used frameworks for web application and API creation Node.js in conjunction with Express. Figure 8 presents the API working architecture. The API routes the requests to the appropriate service in the back-end.

The API serves as a communication hub, facilitating seamless data exchange and interaction between various components of the system. The left side of the diagram represents a client’s side, where users interact with the system through two primary interfaces:Web Application: Users can access and control various features of the wheelchair through a dedicated web application. This interface allows users to input commands, view sensor data, and manage their preferences.Chatbot: The chatbot provides a conversational interface for users to interact with the system using natural language. It processes user requests, translates them into commands, and relays them to the API.

The API sits at the center of the architecture, acting as an intermediary between the client-side interfaces and the back-end services responsible for processing data and controlling the wheelchair. It receives requests from the web application and chatbot, routes them to the appropriate services and returns responses to the clients. The right side of the diagram depicts the back-end services responsible for executing specific tasks based on requests received from the API. These services include

Database: Stores user preferences, including information like speed settings, frequent destinations, and potentially other personalized data.Facial Recognition: Processes images from the user camera to perform facial recognition, enabling user authentication and personalized features.Navigation: Handles a wheelchair’s autonomous navigation, utilizing data from sensors like LiDAR and stereo camera to plan paths, avoid obstacles, and reach desired destinations.Other Services: The diagram also indicates the potential for integrating additional back-end services as needed to expand the system’s functionality.

The arrows in the diagram represent the flow of data and communication between the different components: (i) clients (web application and chatbot) send requests to the API; (ii) the API routes these requests to the appropriate back-end services; (iii) back-end services process the requests, access necessary data (e.g., from the database), and perform their designated functions; (iv) services send responses back to the API; and, (v) the API relays these responses to the clients, providing feedback or updates to the users.

### 3.6. Interaction Between Hardware and API

Figure 9 presents the interaction between the hardware and API system module. The hardware components provide the sensory input and actuation mechanisms for the wheelchair system. On the other hand, the API plays a crucial role in coordinating communication between the hardware, the web interface, and the chatbot, enabling control and data exchange.

The ZED camera captures images and generates depth maps in real-time, contributing to visual odometry, 3D mapping, and object recognition. The LiDAR enables obstacle detection and environment mapping, providing crucial data for navigation. The DC voltage sensor monitors the battery charge level of the wheelchair.

The Jetson serves as the central processing unit for the wheelchair, handling tasks related to autonomous navigation, sensor data processing, and communication with the API. It collects data from the various sensors, including LiDAR, camera, and DC voltage sensor transmitted to the API via ROS topics, using the Rosbridge library for communication between ROS and the web interface.

The API processes the sensor data, potentially performing tasks like interpreting LiDAR data for obstacle avoidance, analyzing camera images for facial recognition, or retrieving user preferences from the database. It relays commands from the web interface or chatbot to the Jetson AGX Orin, controlling the wheelchair’s movement, speed, and other functionalities. The tablet serves as the user interface for controlling the wheelchair and also facilitates facial recognition.

### 3.7. Speed Control

We developed three levels of wheelchair speed—high, medium, and low—with speeds of 0.55 m/s, 0.33 m/s, and 0.20 m/s, respectively. This classification allows adjusting the speed of the wheelchair according to the user’s demand level or specific condition, providing safe and efficient navigation.

### 3.8. Facial Recognition

To create the facial recognition system, we developed a user interface using the Python programming language. It is possible to add and remove users as well as to add parameters, such as an administrator or regular user and wheelchair speed, where the administrator can add or change all the user parameters.

### 3.9. People Groups’ Recognition

To enhance human–robot interactions, our system has incorporated the Robot-Centric Group Detection and Tracking System (REGROUP) into our proposed framework. This cutting-edge system empowers robots to effectively perceive and interpret team dynamics by detecting and tracking human groups from an ego-centric perspective. Utilizing a crowd-aware, tracking-by-detection approach, REGROUP leverages advanced person re-identification deep learning features to resolve group data association challenges. The system is designed to be resilient to common real-world visual challenges, including occlusion, camera motion, shadows, and varying lighting conditions. Furthermore, REGROUP operates in real time, ensuring practical applicability in dynamic environments [22].

### 3.10. Security and Privacy

We developed the system following the Brazilian General Data Protection Act (LGPD) to further strengthen privacy protections and align with legal requirements in Brazil. The LGPD establishes guidelines for the collection, processing, and storage of personal data, emphasizing user consent, data security, and transparency.

By incorporating these security measures and adhering to the LGPD principles, we prioritize user privacy and data security, which is crucial not just for legal compliance but also for building user trust, especially considering the sensitive nature of assistive technologies.

We encrypted data during transmission and storage to minimize the risk of unauthorized access. This includes utilizing encryption algorithms and secure storage solutions to protect user data. We developed an authentication mechanism to restrict access to sensitive user data. The system uses a privacy-preserving facial recognition technique.

## 4. Results

This section presents the results of the implementation of the interactive system. The navigation map was first constructed virtually to enable the initiation of the HRI process through the chatbot. The user initiates interaction with the chatbot via the web application by typing or verbally specifying the desired destination. The chatbot processes this input, confirms the destination verbally, and initiates the autonomous movement of the wheelchair (NARA).

For testing purposes, it was carried out at the facilities of the Federal Institute of Bahia—Campus Vitória da Conquista, in Block G. To visualize the results of the trajectory and system interaction, we used Gazebo/RViz, a web page, images from the Camera (ZED 2i), and an external camera. The first route was from the entrance door of block G to G2. After that, it was requested that the chair go from G2 to G3, and the next route was to return to the entrance of Block G. Figure 10 shows the location of the destinations requested by the user located on the RViz map together with images from ZED 2i and the YOLO algorithm.

We conducted tests at the facilities of the Federal Institute of Bahia—Campus Vitória da Conquista, specifically in Block G. We utilized a web page interface, images from the ZED 2i camera, and an external camera to evaluate the trajectory and system interaction, tools such as Gazebo/RViz. The first route began at the entrance of Block G and proceeded to G2. Subsequently, the wheelchair was directed from G2 to G3, with the final route returning to the entrance of Block G. Figure 10 presents the user-specified destinations on the RViz map, accompanied by images captured by the ZED 2i camera and processed using the YOLO algorithm.

Figure 11 presents the verbal interaction between the user and the chatbot. The user thanks NARA for the service and asks about what is “react”. The chatbot answers using the Gemini AI.

To evaluate the functioning process of the chatbot’s interaction with the user, we recorded a video that can be accessed via the following link: https://youtu.be/NYQ3DNYkVbQ, (accessed on 5 February 2025).

In the user interface, we used facial detection to personalize the service provided by the wheelchair. Through the authentication performed in the user menu, the registered user in the database was successfully identified, as demonstrated in Figure 12.

### 4.1. Lessons Learned on Response Time and Latency Perspectives

The response time of a system plays a key role in determining the overall quality of a HRI. Latency, defined as the delay between a user’s input and the system’s response, directly impacts user satisfaction, usability, and the perceived intelligence of the system. In robotics, where real-time interactions are critical, minimizing latency while ensuring accurate responses is essential to creating effective and intuitive interfaces.

After the tests, we identified the following potential sources of latency and how we dealt with them:Network Latency: Communication between the web interface and ROS via Rosbridge, as well as between the web interface, chatbot, and the API. To mitigate this, we created a local network dedicated to reducing the latency caused by external factors such as internet speed variability or external traffic congestion. Efforts were made to reduce the size and frequency of data transmissions between the components. We optimized the API endpoints to handle requests with minimal processing time, ensuring a smooth flow of data between the web interface and the chatbot,Processing Latency: The time taken by the Jetson to process sensor data, perform facial recognition, handle navigation calculations, and respond to user input. To mitigate this, we leveraged the Jetson platform’s GPU capabilities to accelerate computationally intensive tasks, such as real-time image processing and facial recognition. Tasks were parallelized wherever possible using multi-threading to ensure that multiple processes, such as navigation calculations and sensor data fusion, occur simultaneously without bottlenecks. Sensor data pipelines were streamlined to minimize latency between data acquisition, processing, and response generation.Chatbot Latency: The processing time required for the chatbot to interpret voice commands, interact with Google Gemini, and generate responses. To mitigate this, we used a powerful Jetson processor and a high-speed local network. Moreover, we preprocessed voice input data locally on the device to reduce the size and complexity of data sent to the chatbot’s processing pipeline. We optimized the interaction with Google Gemini by batching API requests and reducing unnecessary back-and-forth communication. This minimized the latency introduced by cloud interactions.API Latency: The time it takes for the API to route requests and handle data transfer between the database, chatbot and the web interface. To mitigate this, we optimized the database queries by indexing frequently accessed tables and using prepared statements to minimize execution time. API requests were distributed across multiple servers to prevent overloading any single endpoint and ensure consistent response times.Facial Recognition Latency: The time it takes for the system to detect and identify the user for personalized service. To mitigate this, we optimized facial recognition models using techniques like quantization, pruning, and training to reduce inference times while maintaining accuracy. Image data were preprocessed locally to reduce noise and resize frames to the optimal resolution required by the model, minimizing computational overhead.People Groups’ Recognition: The time it takes the system to detect and track human groups for social navigation. To mitigate this, we developed the dynamic region of interest. The system dynamically focused on regions of interest where human groups were most likely to appear, reducing the need to process the entire camera feed and saving computational resources. Similar to facial recognition, group detection was performed on selected frames or batched together to minimize redundancy and improve processing speed.

### 4.2. Reviewing Similar Systems in Other Industrial Fields

While the proposed system was designed for personalized navigation and control for individuals with mobility limitations, exploring similar command and control systems in other industrial fields, such as autonomous rail-road amphibious robots for railway maintenance, could provide valuable insights for enhancement and optimization.

These types of systems are designed to operate in complex and dynamic environments, often for inspection, maintenance, or repair tasks on railway infrastructure [23]. They must be capable of transitioning between rail and road, and potentially water environments, requiring complex navigation and control strategies. Reviewing these systems could offer insights for the wheelchair system in several areas.

Rail and amphibious robots must navigate diverse environments, similar to how a wheelchair might encounter various indoor and outdoor spaces. Examining their navigation systems could inform improvements to the wheelchair’s autonomous navigation, particularly regarding obstacle avoidance, path planning, and localization.

These robotic systems typically require real-time control and monitoring for effective operation. Comparing their control interfaces and data visualization techniques could lead to enhancements in the wheelchair’s web interface and data display.

Moreover, railroad robots often use a combination of sensors, including LiDAR, cameras, and inertial measurement units (IMUs), to perceive their surroundings [23]. Comparing the sensor fusion methods and data interpretation techniques could help improve the wheelchair’s ability to understand and react to its environment.

Many industrial robots are remotely operated or monitored through telemetry. Analyzing these remote operation methods and protocols could potentially offer improved user control and monitoring capabilities for the wheelchair system.

Rail maintenance robots operate in high-risk environments, requiring high levels of safety and reliability. Examining their safety protocols, error handling, and redundancy mechanisms could inform best practices for safety in autonomous wheelchair navigation.

These systems generate large amounts of data that are critical for maintenance and reporting. Reviewing how the rail systems handle data management and analysis could help improve the data handling and diagnostic capabilities of the wheelchair system. Both systems aim to optimize autonomy and interactions in their respective domains. However, the railroad system prioritizes task-oriented functionality, while the wheelchair framework prioritizes seamless interactions and user-centric design.

### 4.3. Threats to Validity

This section presents the potential threats to validity, acknowledging the limitations and areas for further investigation. While this research demonstrates promising results in integrating multimodal interactions, real-world adaptability, and personalization for an autonomous wheelchair system, it is essential to address factors that might influence the generalizability and robustness of the conclusions drawn. For this reason, we highlight the following threats:Limited Testing Environment: The testing was conducted in a specific location (Block G of the Federal Institute of Bahia). This limited scope raises questions about the system’s performance and adaptability in diverse environments with varying layouts, obstacles, and user densities. Further testing in different settings is crucial to assess the wheelchair’s generalizability and robustness.Dependence on Pre-mapped Environments: The system relies on pre-existing maps for autonomous navigation. This approach may hinder usability in unfamiliar environments where mapping data are unavailable. Exploring methods for on-the-fly mapping or integration with real-time mapping services could enhance the system’s adaptability.Single User Focus: The testing primarily demonstrates an interaction with a single user. It remains unclear how the system would handle multiple users requesting navigation simultaneously, especially in crowded or shared spaces. Addressing multi-user scenarios and potential conflicts is crucial for broader usability and real-world deployment.Reliance on Internet Connectivity: The chatbot’s integration with Google Gemini suggests a reliance on internet connectivity for certain functionalities. This dependence could pose limitations in areas with poor or intermittent internet access. Exploring offline capabilities or alternative communication methods could improve system reliability.Lack of Quantitative Performance Data: While the sources showcase successful navigation instances, they lack detailed quantitative data on metrics such as navigation accuracy, time efficiency, or user satisfaction across different scenarios. Providing concrete performance metrics would strengthen the evaluation and provide a clearer understanding of the system’s capabilities and limitations.Limited interaction complexity: The tests mainly focused on basic navigation commands and did not explore the system’s response to more intricate, multifaceted user requests or scenarios. The chatbot primarily understands basic navigation commands. Other types of more complex requests rely on Google Gemini for general information. This could lead to inconsistencies in the responses and is not specifically tailored for the robotic system.

## 5. Conclusions

Based on the evaluation of the results obtained, the developed interactive system showed indications of effectiveness in the interaction between users and the autonomous wheelchair, providing an intuitive and personalized control experience. The integration of advanced technologies, such as facial recognition, the integrated Gemini chatbot, and autonomous navigation, demonstrated the ability of the system to provide personalized and responsive services.

The system leverages multiple modes of interaction, combining a web interface, chatbot communication, and facial recognition. This approach enhances the user’s control experience by offering flexibility and accommodating diverse preferences and abilities.

The wheelchair’s integration with REGROUP, a robot-centric group detection and tracking system, allows it to operate effectively in dynamic, real-world environments. The ability of REGROUP to handle occlusion, camera movement, and varying lighting conditions ensures that the wheelchair can navigate safely and efficiently even in complex social settings.

The system incorporates user preferences, such as speed and frequent destinations, stored in a database accessed through an API. This personalization capability, coupled with facial recognition for user identification, tailors the wheelchair’s behavior to individual needs, improving user satisfaction, and overall system usability.

In our future work, we intend to integrate a login system into the chatbot in a web interface to provide a more personalized and secure experience to users based on the information registered in the database and route redirection, restricting access to certain website features based on the user type (admin or user), in addition to improving our autonomous navigation system by recognizing groups of people. An instant-time localization system [24], previously implemented, will be incorporated into the proposal. For this reason, we propose the following roadmap:Expanding Chatbot Language Support: Implementing multilingual capabilities to cater to a broader range of users, enhancing accessibility and inclusivity.Addressing Localization Challenges: Adapting the system to operate efficiently in diverse environments by incorporating regional mapping, cultural preferences, and context-specific interactions.Adapting to Different Hardware Platforms: Ensuring compatibility with a variety of hardware configurations, including lower-cost alternatives, to improve scalability and accessibility.Enhanced Social Interaction Capabilities: Developing advanced algorithms to improve group detection and interaction, enabling robots to better navigate and communicate in social contexts.Integration of Real-Time Localization Systems: Incorporating instant-time localization technologies to enhance navigation precision and reliability.Develop a More Robust Command Set: Expanding the command set of the chatbot to handle a wider range of user requests, including adjustments to the wheelchair’s speed, monitoring of sensor data, and activation of system functionalities. This would require more sophisticated natural language understanding capabilities.

## Figures and Tables

**Figure 1 sensors-25-00987-f001:**
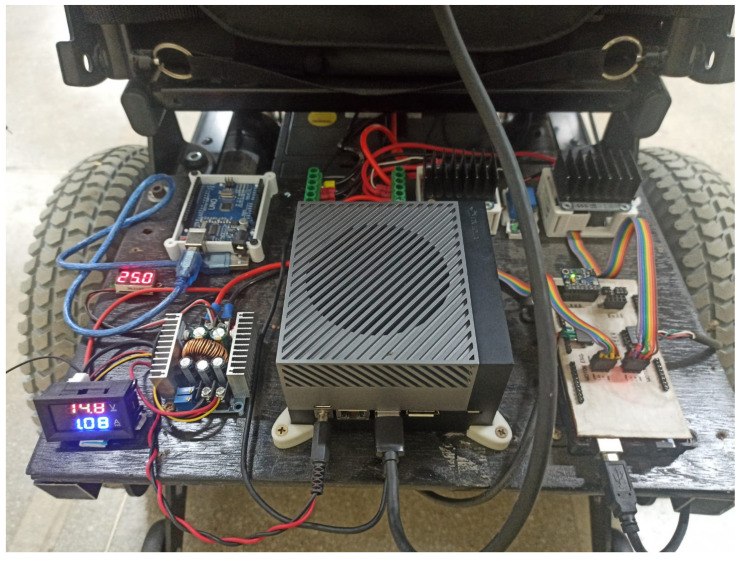
Hardware used in the wheelchair.

**Figure 2 sensors-25-00987-f002:**
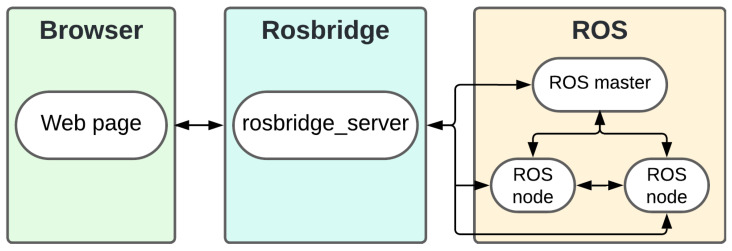
Rosbridge communication diagram between the web interface and ROS.

**Figure 3 sensors-25-00987-f003:**
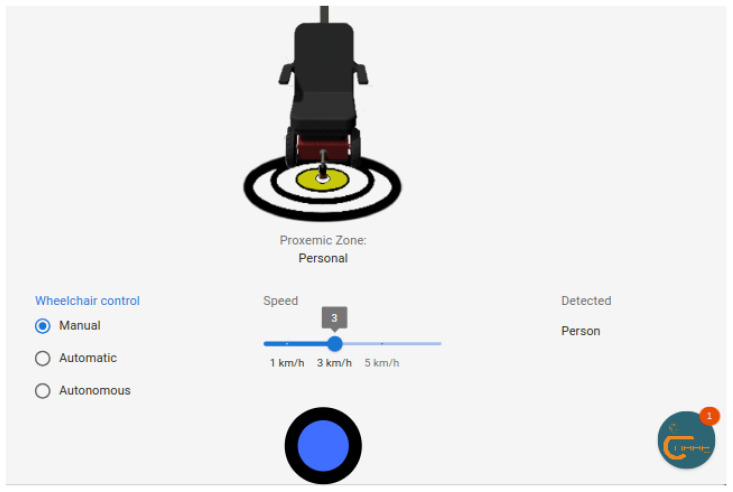
Manual control of the wheelchair using a Joystick.

**Figure 4 sensors-25-00987-f004:**
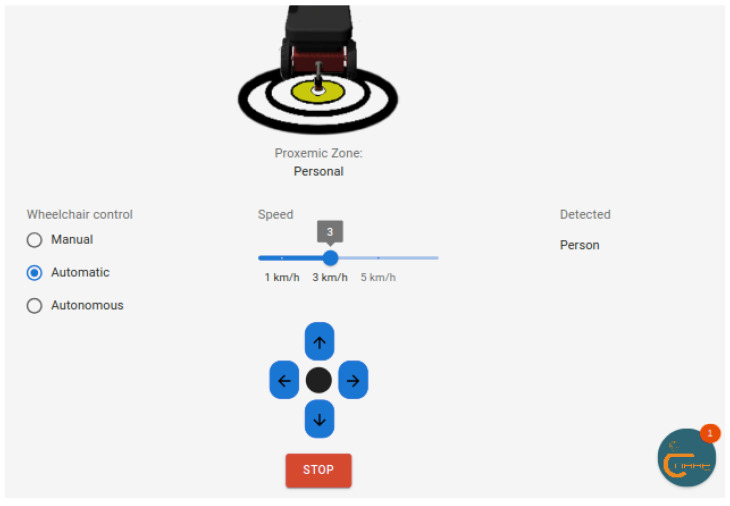
Automatic wheelchair control using directional arrows.

**Figure 5 sensors-25-00987-f005:**
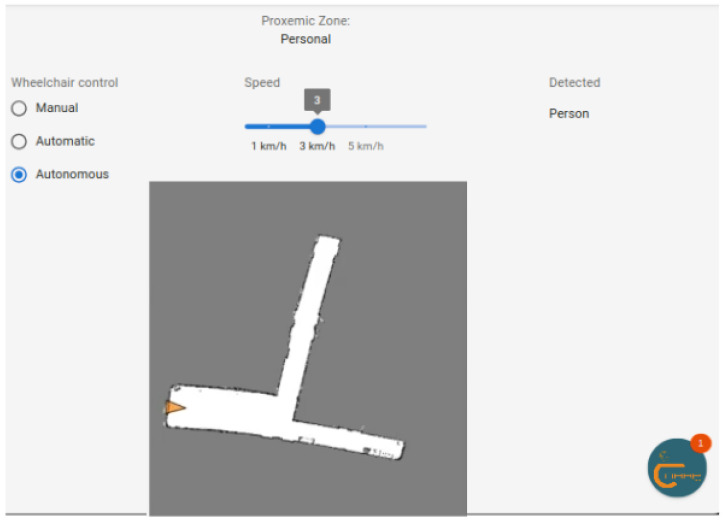
Autonomous wheelchair control.

**Figure 6 sensors-25-00987-f006:**
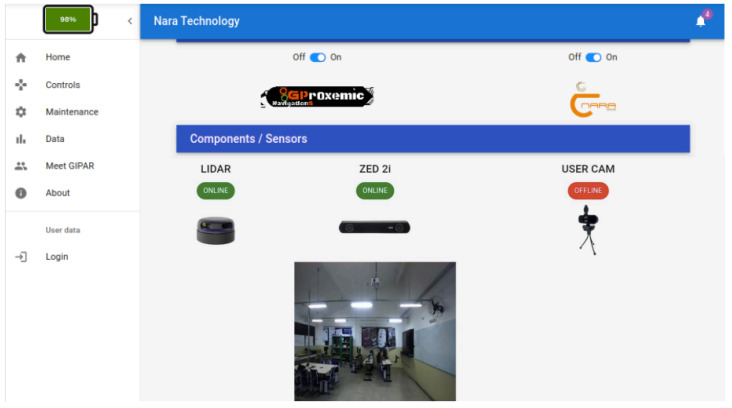
Data page.

**Figure 7 sensors-25-00987-f007:**
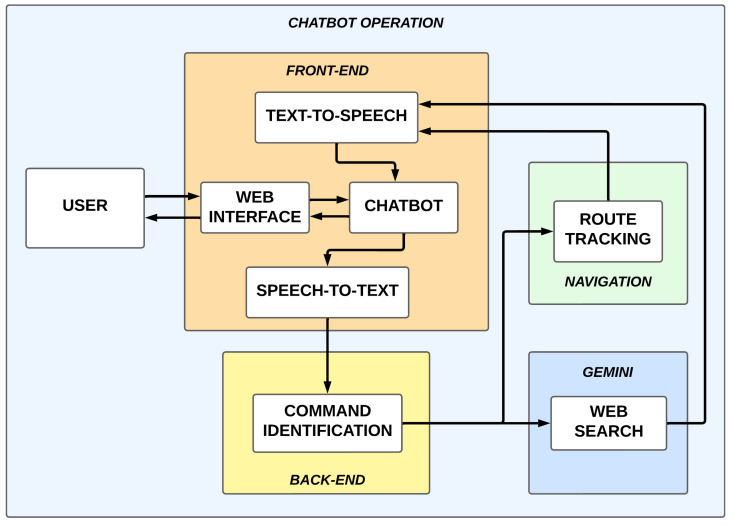
Chatbot working diagram.

**Figure 8 sensors-25-00987-f008:**
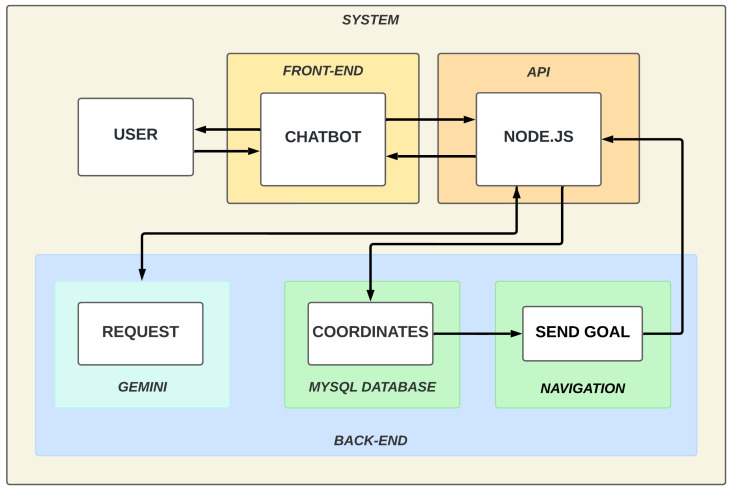
API working architecture.

**Figure 9 sensors-25-00987-f009:**
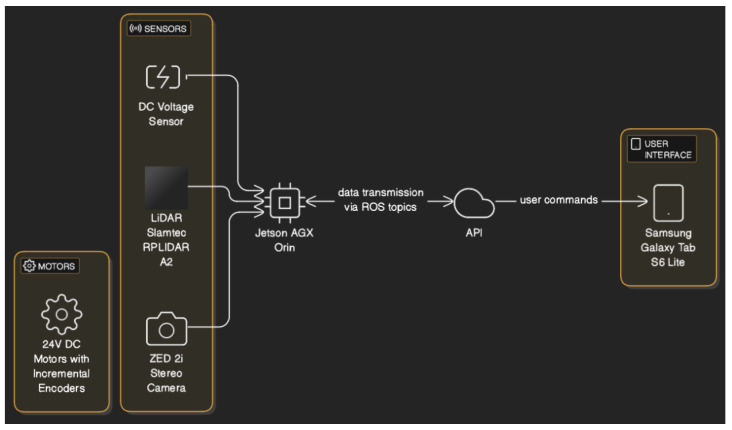
Interaction between the hardware and API.

**Figure 10 sensors-25-00987-f010:**
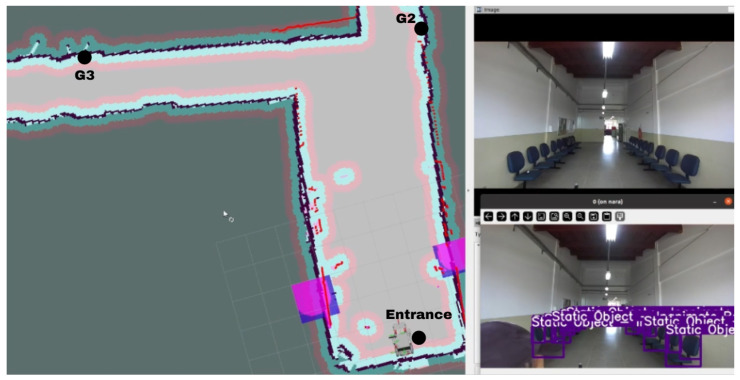
Wheelchair trajectory.

**Figure 11 sensors-25-00987-f011:**
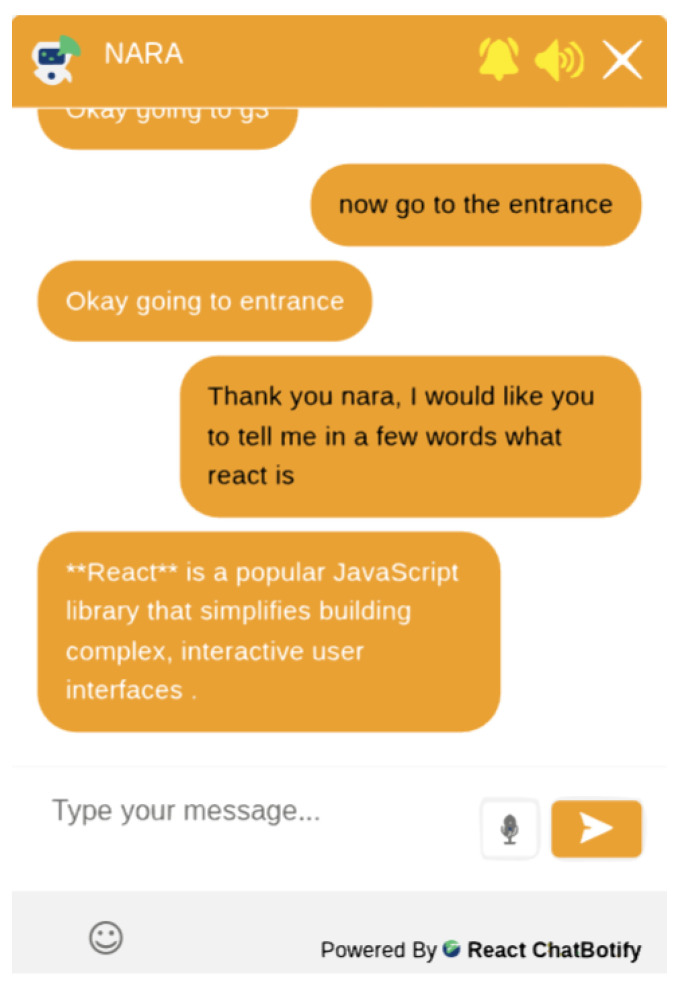
Interaction between the user and the chatbot with integrated Google Gemini.

**Figure 12 sensors-25-00987-f012:**
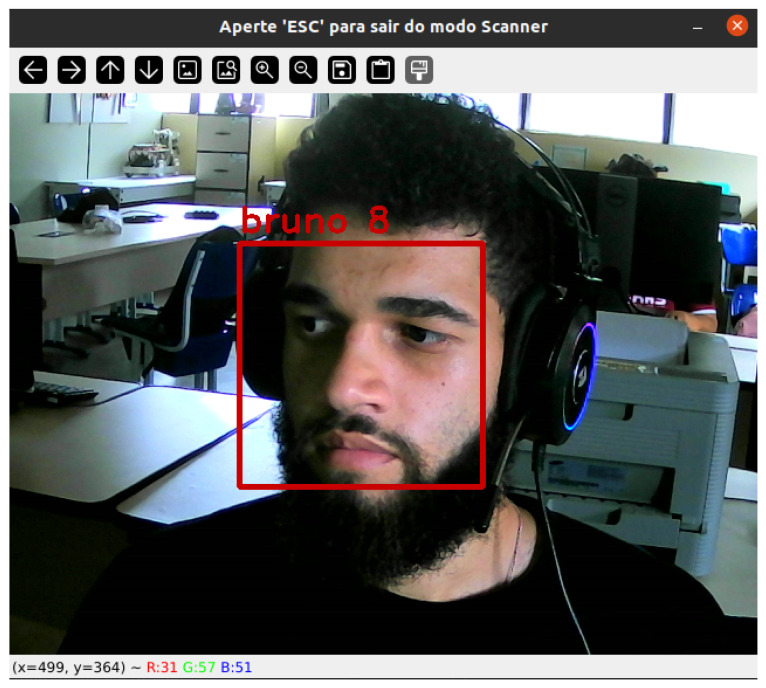
Facial recognition.

## Data Availability

Data are contained within the article.

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
