# Peer review of "Design and Implementation of an Interactive System for Service Robot Control and Monitoring"

_sensors, 2025, doi:10.3390/s25040987_

Round 1

Reviewer 1 Report

Comments and Suggestions for Authors

it is attached. 

Author Response

1: The manuscript includes relevant prior works but does not provide a detailed comparative analysis. It would benefit from more clearly articulating how its contributions stand out compared to existing studies. Furthermore, the discussion on advancements in HRI is somewhat superficial. Expanding on user experience and ethical considerations in robotics would provide better context. To enhance the literature review, the authors are advised to reference the following paper: 10.1007/s12541-019-00018-y.  We improved the discussion with a detailed comparative analysis. We described how our contribution stands out compared to existing studies; 2.-The results section lacks sufficient quantitative performance metrics. Including measures such as the accuracy of facial recognition, the success rate of voice commands, or user satisfaction scores would make the findings more robust. Additionally, the paper could benefit from discussing the limitations encountered during testing, as this would provide insights into scalability and broader usability. We sincerely appreciate your observations and agree on the importance of including quantitative metrics in our study. However, in this initial version, we focused on establishing the methodology and testing key components in a controlled environment. We plan to conduct tests in real-world settings to gather quantitative data, including facial recognition accuracy and voice command success rates, as well as analyzing encountered limitations. We aim to incorporate these aspects into a future version of the paper to broaden our analysis and meet the raised expectations. 3.- There are several grammatical errors and formatting inconsistencies (e.g., abrupt paragraph transitions, irregular page numbering) that detract from the manuscript’s overall quality. Additionally, some sections rely heavily on technical jargon, which may limit the accessibility of the paper to non-technical readers or stakeholders. Simplifying these sections could improve readability. We review the paper's language and presentation to provide a better reading experience. 4.- The manuscript does not sufficiently address potential security and privacy risks, particularly those related to storing user preferences and performing facial recognition. Including a discussion on how these risks are mitigated would strengthen the manuscript’s credibility and completeness. The Federal Insitute of Bahia is an educational institution established by the Brazilian government. For this reason, our projects follow the Brazilian General Data Protection Law (LGPD). 5.- The future directions section, while present, is somewhat vague. More specific plans are needed, such as expanding the chatbot’s language support, addressing localization challenges, and adapting the system to different hardware platforms. A detailed roadmap would provide a clearer vision for the project’s potential. We included a roadmap to guide future work and provide a clearer vision for the project's portential. 6.- The discussion on how this work advances existing research is limited. A deeper analysis of its contributions to areas like multimodal interaction and real-world adaptability would provide greater context and impact. We improved the discussion on how this work advances. Please verify Related works, Results, and Conclusion sections. 7.-Incorporating benchmarks or comparative tests would help highlight the system’s effectiveness. Metrics such as improvements in navigation efficiency, responsiveness, or user satisfaction would provide more compelling evidence of the system’s success. Please verify answer 2,  8.-Ethical and privacy concerns, especially regarding data storage and user authentication, require more attention. A thorough exploration of these issues would add depth to the discussion and address potential concerns from users and stakeholders. Please verify answer 4.  9.- The manuscript would benefit from a detailed revision to correct grammatical issues and improve overall clarity. Simplifying technical explanations where possible would make the work more accessible to a wider audience. Please verify answer 3. 10. Future work should be clearly outlined with specific, actionable steps. For example, the authors could detail plans for addressing the system’s limitations, expanding its capabilities, and adapting it to diverse scenarios. Please verify answer 5.

Reviewer 2 Report

Comments and Suggestions for Authors

This paper presents an interactive control system for a autonomous service robot using ROS framework. Based on the SOTA, the API framework was half- completed and not fully established. The robot demonstration and performance following the software command and control are poorly showed in the paper. It is recommended to be a resubmission after a major improvement on both experimental testing and paper illustration. Several major comments and concerns are listed below.

1. The hardware deployment sytem are expected to be illustrated with a systematic framework graph or picture. And the interaction between the hardware and API system module should be included.

2. Figures and tables illustration are not informative. Figure 3 is not showing any useful information related to energy saving or robot endurance. Figure 10 only shows a menu which are not connected to the face detection module. Table 1 is just a speed definition, that is not related to the robot practical running performance.

3. The interation between user command and robot was not demonstrated with practical robot demonstration and verification. Figure 12 is only a chat. Dose the robot actually deploy the command?

4. The indoor vehicle movability performance highly relies on the SLAM performance while running. The close-loop localization accuracy and collision avoidance performance analysis are not presented.

5. Please do the carefully grammar changes and check for spelling and grammar mistakes. For instance, in abstract, “em- ploys” does not need dash mark. 

Comments on the Quality of English Language

The paper is not well structured and quantitative analysis is not enough. The writing should be improved based on this.

Author Response

1.- The hardware deployment sytem are expected to be illustrated with a systematic framework graph or picture. And the interaction between the hardware and API system module should be included. We created a new section and Figure 9 to illustrate the deployment system and the interaction between the hardware and API. 2.- Figures and tables illustration are not informative. Figure 3 is not showing any useful information related to energy saving or robot endurance. Figure 10 only shows a menu which are not connected to the face detection module. Table 1 is just a speed definition, that is not related to the robot practical running performance. We agree with the suggestions. We deleted the Figure 3, Figure 10, and Table 1,  3.- The interation between user command and robot was not demonstrated with practical robot demonstration and verification. Figure 12 is only a chat. Dose the robot actually deploy the command?. We demonstrated the interaction between the user command and the robot in the following YouTube video: https://youtu.be/NYQ3DNYkVbQ. Figure 12 presents a conversation between the user and the chatbot. The robot deploys the command, as can be seen in the video demonstration. 4.- The indoor vehicle movability performance highly relies on the SLAM performance while running. The close-loop localization accuracy and collision avoidance performance analysis are not presented. The paper focuses on the methodology of human-machine interaction of a wheelchair and the user, for this reason, complementary requirements to it, such as the SLAM case, were not evaluated. Details of the implemented SLAM, please see https://ieeexplore.ieee.org/document/9605464. This reference was added to the paper in section 3.4 .   5.- Please do the carefully grammar changes and check for spelling and grammar mistakes. For instance, in abstract, “em- ploys” does not need dash mark.  We review the paper's language and presentation to provide a better reading experience.

Round 2

Reviewer 1 Report

Comments and Suggestions for Authors

The revised paper has been improved significantly and it has addressed all the raised concerns. It is highly recommended for publication at the current version.

Author Response

Response to reviewer’s comments

We would like to sincerely thank you and the reviewers for the time and effort spent in evaluating our manuscript. We found the comments from the reviewers to be highly constructive and encouraging, and we think they have immensely improved the quality of the paper.

Our paper is revised to a substantial degree. In addition to revising based on reviewers comments, we have also clarified, edited, moved, removed, and added additional text. 

For clarity, we have provided a detailed point-by-point response to each reviewer’s comment. In the response document:

  • We have included references to the sections where the changes have been made in the manuscript.
  • The corresponding changes made to the manuscript are shown here in red text for easy reference.

Paragraphs or sections which are significantly new are also marked in red in the manuscript, although many other edits have also been conducted.

Comment#1: Literature Review Gaps

The manuscript includes relevant prior works but does not provide a detailed comparative analysis. It would benefit from more clearly articulating how its contributions stand out compared to existing studies. Furthermore, the discussion on advancements in HRI is somewhat superficial. Expanding on user experience and ethical considerations in robotics would provide better context. To enhance the literature review, the authors are advised to reference the following paper: 10.1007/s12541-019-00018-y.

Authors’ Response:

Thank you for these suggestions, We improved the discussion with a detailed comparative analysis. We described how our contribution stands out compared to existing studies;

Most of Section 2 was revised, see red text.

We improved the discussion on advancements in HRI (see red text in Section 1) and included the suggested reference. 

Comment#2: Evaluation Metrics

The results section lacks sufficient quantitative performance metrics. Including measures such as the accuracy of facial recognition, the success rate of voice commands, or user satisfaction scores would make the findings more robust. Additionally, the paper could benefit from discussing the limitations encountered during testing, as this would provide insights into scalability and broader usability.

Authors’ Response:

We sincerely appreciate your observations and agree on the importance of including quantitative metrics in our study. However, in this initial version, we focused on establishing the methodology and testing key components in a controlled environment. We plan to conduct tests in real-world settings to gather quantitative data, including facial recognition accuracy and voice command success rates, as well as analyzing encountered limitations. We aim to incorporate these aspects into a future version of the paper to broaden our analysis and meet the raised expectations.

Moreover, we created threats to the validity section to discuss the limitations encountered during testing and the lack of quantitative performance data.

Section 4.1 is new.

Comment#3: Language and Presentation

There are several grammatical errors and formatting inconsistencies (e.g., abrupt paragraph transitions, irregular page numbering) that detract from the manuscript’s overall quality. Additionally, some sections rely heavily on technical jargon, which may limit the accessibility of the paper to non-technical readers or stakeholders. Simplifying these sections could improve readability.

Authors’ Response:

Thank you for pointing out all of these small errors, we have done our best to address them. We reviewed the paper's language and presentation to provide a better reading experience.

Comment#4: Security Concerns

The manuscript does not sufficiently address potential security and privacy risks, particularly those related to storing user preferences and performing facial recognition. Including a discussion on how these risks are mitigated would strengthen the manuscript’s credibility and completeness.

Authors’ Response:

We appreciate your comment. The Federal Institute of Bahia is an educational institution established by the Brazilian government. For this reason, our projects follow the Brazilian General Data Protection Law (LGPD).

We created a security and privacy section to discuss the risks and how we deal with them.

Section 3.10 is new.

Comment#5: Future work

The future directions section, while present, is somewhat vague. More specific plans are needed, such as expanding the chatbot’s language support, addressing localization challenges, and adapting the system to different hardware platforms. A detailed roadmap would provide a clearer vision for the project’s potential.

Authors’ Response:

Thank you for your observation. We included a roadmap to guide future work and provide a clearer vision for the project's potential.

Most of Section 5 is new or revised, see red text.

Comment#6: Enhanced Comparative Analysis

The discussion on how this work advances existing research is limited. A deeper analysis of its contributions to areas like multimodal interaction and real-world adaptability would provide greater context and impact.

Authors’ Response:

Thank you for this feedback. For this reason, we improved the discussion on how this work advances. 

Please verify Section 2, Section 4, and Section 5.

Comment#7: Inclusion of Quantitative Metrics

Incorporating benchmarks or comparative tests would help highlight the system’s effectiveness. Metrics such as improvements in navigation efficiency, responsiveness, or user satisfaction would provide more compelling evidence of the system’s success.

Authors’ Response:

Please verify the answer to Comment#2

Comment#8: Ethical and Security Considerations

Ethical and privacy concerns, especially regarding data storage and user authentication, require more attention. A thorough exploration of these issues would add depth to the discussion and address potential concerns from users and stakeholders.

Authors’ Response:

Please verify the answer to Comment#4.

Comment#9: Language Refinement

The manuscript would benefit from a detailed revision to correct grammatical issues and improve overall clarity. Simplifying technical explanations where possible would make the work more accessible to a wider audience.

Authors’ Response:

Please verify the answer to Comment#3.

Comment#10: Detailed Future Directions

Future work should be clearly outlined with specific, actionable steps. For example, the authors could detail plans for addressing the system’s limitations, expanding its capabilities, and adapting it to diverse scenarios.

Authors’ Response:

Please verify the answer to Comment#5

Reviewer 2 Report

Comments and Suggestions for Authors

This paper is the second time submission after improvement following previous reviewing comments. It is much better in terms of presentation and quality of result analysis.

By the statement of authors: this paper focuses on the interaction software between the human and the robot. In this case, I am concerned about the response and latency perspectives of the interaction. Can the author show any quantitative analysis graph or profile? And, in the current status, has the testing investigated the limits of the system in handling complex interactions?

In addition, the author should review the similar command and control system on the other industrial field, like the autonomous rail-road amphibious robotic system for railway maintenance.

I recommend a minor revision after addressing above comments.

Author Response

Response to reviewer’s comments

We would like to sincerely thank you and the reviewers for the time and effort spent in evaluating our manuscript. We found the comments from the reviewers to be highly constructive and encouraging, and we think they have immensely improved the quality of the paper.

Our paper is revised to a substantial degree. In addition to revising based on reviewers comments, we have also clarified, edited, moved, removed, and added additional text. 

For clarity, we have provided a detailed point-by-point response to each reviewer’s comment. In the response document:

  • We have included references to the sections where the changes have been made in the manuscript.
  • The corresponding changes made to the manuscript are shown here in red text for easy reference.
  • Paragraphs or sections which are significantly new are also marked in red in the manuscript, although many other edits have also been conducted.

Comment#1: Hardware

The hardware deployment system is expected to be illustrated with a systematic framework graph or picture. And the interaction between the hardware and API system module should be included.

Authors’ Response:

Thank you for your observation. We improved Section 3.5 and created a new Section 3.6 and Figure 9 to illustrate the deployment system and the interaction between the hardware and API.

Comment#2: Figures

Figures and tables illustrations are not informative. Figure 3 is not showing any useful information related to energy saving or robot endurance. Figure 10 only shows a menu which is not connected to the face detection module. Table 1 is just a speed definition, that is not related to the robot practical running performance.

Authors’ Response:

We agree with the suggestions. We deleted Figure 3, Figure 10, and Table 1.

Comment#3: Interaction

The interaction between user command and robot was not demonstrated with practical robot demonstration and verification. Figure 12 is only a chat. Does the robot actually deploy the command?

Authors’ Response:

We demonstrated the interaction between the user command and the robot in the following YouTube video: https://youtu.be/NYQ3DNYkVbQ

Figure 12 presents a conversation between the user and the chatbot. The robot deploys the command, as can be seen in the video demonstration.

Comment#4: Slam

The indoor vehicle movability performance highly relies on the SLAM performance while running. The close-loop localization accuracy and collision avoidance performance analysis are not presented.

Authors’ Response:

The paper focuses on the methodology of human-machine interaction of a wheelchair and the user, for this reason, complementary requirements to it, such as the SLAM case, were not evaluated. Details of the implemented SLAM, please see https://ieeexplore.ieee.org/document/9605464. This reference was added to the paper in Section 3.4

Comment#5: grammar

Please do the grammar changes carefully and check for spelling and grammar mistakes. For instance, in the abstract, “em- ploys” does not need a dash mark.

Authors’ Response:

Thank you for pointing out all of these small errors, we have done our best to address them. We reviewed the paper's language and presentation to provide a better reading experience.

Comment#6: Final

The paper is not well structured and quantitative analysis is not enough. The writing should be improved based on this.

Authors’ Response:

We reviewed the paper's language and presentation for a better reading experience. We also performed improvements suggested by Reviewer#1. We provided a discussion with a detailed comparative analysis. We agree on the importance of improving the quantitative metrics. For this reason, we created threats to the validity section.
